# Identifying Modelling Issues through the Use of an Open Real-World Flood Dataset

**Vasilis Bellos** [1,*] **, Ioannis Kourtis** [2] **, Eirini Raptaki** [2] **, Spyros Handrinos** [3] **, John Kalogiros** [4] **, Ioannis A. Sibetheros** [5] **and Vassilios A. Tsihrintzis** [2]

1    Laboratory of Ecological Engineering and Technology, Department of Environmental Engineering, Democritus University of Thrace, 12 Vas. Sofias St., 67100 Xanthi, Greece
2    Centre for the Assessment of Natural Hazards and Proactive Planning, Laboratory of Reclamation Works and Water Resources Management, School of Rural, Surveying and Geoinformatics Engineering, National Technical University of Athens, 9 Heroon Polytechniou St., Zografou, 15780 Athens, Greece
3    MSc Civil Engineering Program, Department of Civil Engineering and Geosciences, Delft University of Technology, Stevinweg 1, 2628 Delft, The Netherlands
4    Institute for Environmental Research and Sustainable Development, National Observatory of Athens, I. Metaxas & Vassilis Pavlou St., Old Penteli, 15236 Athens, Greece
5    Department of Civil Engineering, School of Engineering, University of West Attica, 250 Petrou Ralli & Thivon St., Egaleo, 12244 Athens, Greece
*    Correspondence: vbellos@env.duth.gr

**Abstract:** The present work deals with the reconstruction of the flood wave that hit Mandra town (Athens, Greece) on 15 November 2017, using the framework of forensic hydrology. The flash flood event was caused by a huge storm event with a high level of spatial and temporal variability, which was part of the Medicane Numa-Zenon. The reconstruction included: (a) the post-event collection of 44 maximum water depth traces in the town; and (b) the hydrodynamic simulation employing the HEC-RAS and MIKE FLOOD software. The derived open dataset (which also includes additional data required for hydrodynamic modeling) is shared with the community for possible use as a benchmark case for flood model developers. With regards to the modeling issues, we investigate the calibration strategies in computationally demanding cases, and test whether the calibrated parameters can be blindly transferred to another simulator (informed modeling). Regarding the calibration, it seems that the coupling of an initial screening phase with a simple grid-search algorithm is efficient. On the other hand, the informed modeling concept does not work for our study area: every numerical model has its own dynamics while the parameters are of grey-box nature. As a result, the modeler should always be skeptical about their global use.

**Keywords:** forensic hydrology; flood modeling; open dataset; HEC-RAS; MIKE FLOOD





## 1. Introduction

Floods occur in both rural and urban environments and are among the most destructive natural hazards, creating huge economic losses and casualties at global scale [1,2]. Impervious surfaces pose a major effect on watershed hydrology [2], since urban sprawl affects total runoff volumes, peak flow rates, and catchment response times. Moreover, discharges, associated with storm events with high and/or low probability of occurrence before development, increase after urban development takes place [3]. The combined effects of urban development and climatic variability may affect the urban water cycle [1]. Moreover, extreme urbanization and the projected climate change have led the scientific community to focus more on urban flood risk, urban flood dynamics and flood mitigation measures, estimation of return periods of extreme events through extreme value analysis, in both quantitative and qualitative terms, and the update of intensity–duration–frequency curves [4–12].

In our era, physically based simulators are the main tools used for simulating the flow dynamics driven by urbanization, land use/land cover (LULC) changes, and climatic variability in the urban environment [7]. The main approach employed is the 1D approach. However, 1D-1D, 1D-2D, and 2D approaches are also applied [13]. Various uncertainties, both aleatory and epistemic, are associated with these models and this is mainly because of the model structure, initial and boundary conditions, input and forcing data. Moreover, the rational estimation of the various parameters, incorporated in these models, requires a calibration procedure [14]. For example, parameters regarding the proper coupling of 1D flow in the sewer system with 2D flow on the surface of the catchment include the proper representation of buildings and other obstacles to the flow, the estimation of mass losses due to infiltration or interception, or the estimation of energy losses due to friction, among others [14].

Accurate simulation of flood events is extremely difficult. The main reason for this is associated with the absence of a proper dataset which can be used for model calibration-validation purposes, especially in urban catchments which are partially gauged or completely ungauged. It should be noted herein that even in a gauged urban catchment the monitoring system is often destroyed during a flood. The framework of forensic hydrology, as proposed by [15], which comprises five steps (i.e., information gathering; hydrometeorological and hydrological analysis; hydrodynamic analysis; integrative analysis; and final diagnosis), pays special attention to data collection for a proper reconstruction of the event. Some recent studies propose the post-event data collection using some proxies as measurements, which are mainly the maximum flood depths at several observational points [16,17], while other studies are based on data derived by flood event reconstruction using physical modelling [18,19]. Other strategies with potential merit are the crowd sourcing of data derived by social media, also focusing on water depths at specific time moment [20], and finally, remote sensing data, providing flood extents at various time points [21].

This paper is divided in two parts. In the first part, we describe and share a full dataset, deposited in Zenodo platform, acquired using the forensic hydrology framework at the site. The flood dataset is associated with the catastrophic flood event that hit Mandra town in Athens, Greece, on 15 November 2017, causing 24 casualties. The purpose of this open dataset is to provide a complete dataset for benchmarking flood simulators. In the second part, we focus on the second and third steps of forensic hydrology framework. Specifically, we reconstructed the flood event with the assistance of commercial, physically based simulators, and we investigate some practical issues regarding flood modeling, which can be summarized as follows: (a) the strategy for calibrating the required parameters with respect to the computational burden; (b) the potential transferability of the calibrated parameters from one software to the other, in order to identify if the calibrated parameters are of global or grey-box nature.

## 2. Materials and Methods

### 2.1. Study Area

The town of Mandra is located in Attica, in the western part of the greater metropolitan area of Athens, Greece (Figure 1). It is built at the outlet of two catchments, namely Agia Aikaterini and Soures catchments (Figure 1), which are part of the greater river system of Sarantapotamos, and extends along the eastern-southeastern foothills of Mt. Pateras (1130 m).

The storm event under study caused the severe Mandra flood, which occurred between the 14 November 2017 at 23:00 UTC and the 15 November 2017 at 12:00 UTC and was part of the Medicane Numa-Zenon. It was a highly localized phenomenon with extreme spatial and temporal variability. According to the National Observatory of Athens (NOA), which recorded the rainfall field with a mobile X-band polarimetric weather radar (XPOL), the total rainfall on Mt. Pateras, above Nea Peramos and Mandra, exceeded 200 mm in depth during the 6-h main storm event, with instant rainfall intensities reaching peak values

of up to 120–140 mm/h, while the accumulated rainfall in 10 h reached nearly 300 mm (Figure 2, left).

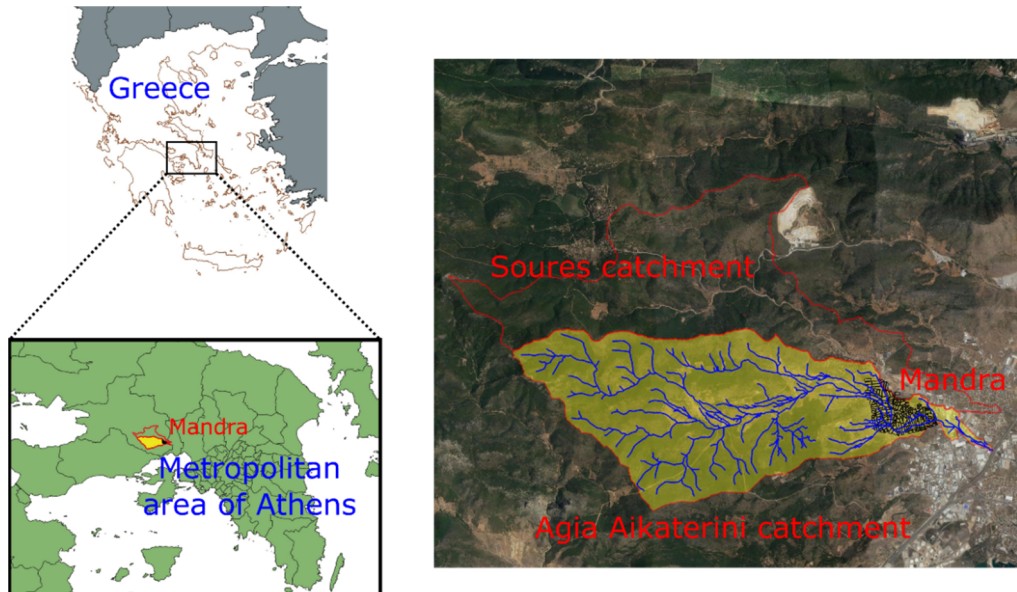

**Figure 1.** General location of Mandra town and the Agia Aikaterini and Soures catchments.

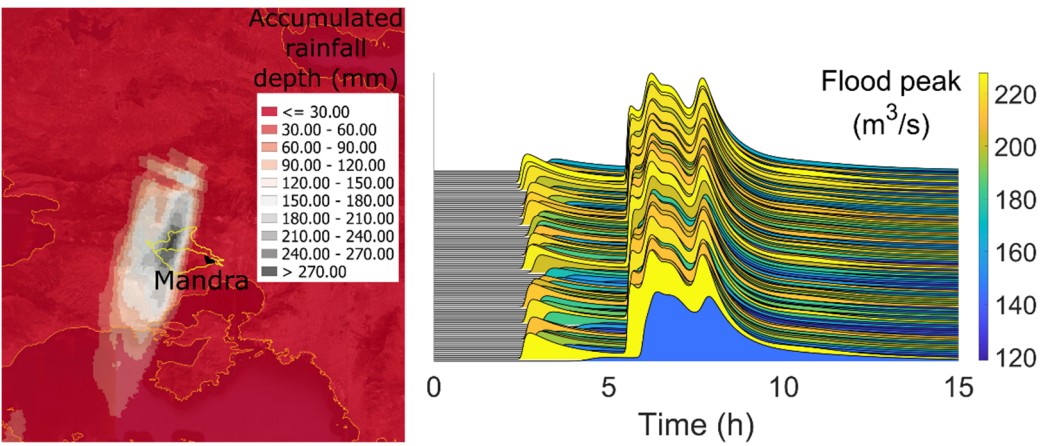

**Figure 2.** Accumulated rainfall depth recorded between 14 November 2017 at 22:30 and 15 November 2017 at 08:30 GMT (left); Joyplot of the 100 input hydrographs computed at Mandra town starting from 14 November 2017 at 22:30 until 15 November 2017 at 13:30 (right).

The response of the storm was a flash flood wave, which hit the town during the night, with catastrophic consequences. It is one of the biggest multi-fatality flood events recorded in Greece. Several researchers reconstructed the event, simulating the flood hydrographs at the inlet of the town [8,22–28].

In this work, the input hydrographs were taken from a previous study [29] which reconstructed the flood dynamics of Agia Aikaterini catchment using a 2D-SWE-based simulator named FLOW-R2D [23]. A parametric and input data uncertainty analysis was performed. Therefore, the output was an ensemble of 100 hydrographs at the outlet of the catchment and at the inlet of the town (Figure 2, right). The uncertainty band of the flood discharge peak ranges between 120 m$^3$/s and 220 m$^3$/s, while the median was about 180 m$^3$/s. This peak is in accordance with the post event rough estimation of the flood peak made by [24].

## 2.2. Data Collection

The research team visited the field four days after the flood event, and specifically, on 19 November 2017. During the visit, they derived a dataset of 44 maximum water depths using the dry mud or leaf footprints on several building walls as a proxy indication. The depth was measured with a measuring tape while the exact coordinates were recorded with a hand-held GPS. Figure 3 depicts a general view of the observation points while Figures 4–7 depict some indicative photos taken in the field. The majority of the observation points are along the two main streets of Mandra, since the major flood impact was observed in these roads. Specifically, the roads are Vaggeli Koropouli str. and Str. Nik. Rokka str., which are the extensions of Agia Aiakterini str (which in fact is the extension of the ephemeral Agia Aikaterini stream).

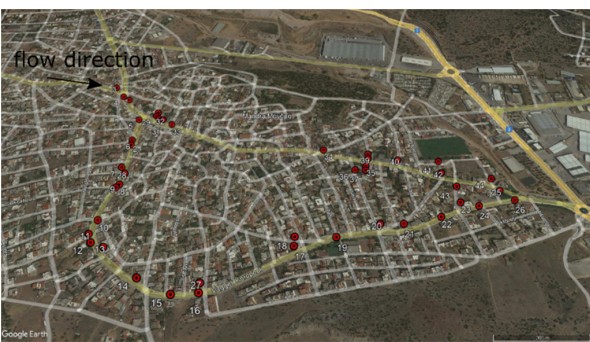

**Figure 3.** Location of observation points across Mandra town.

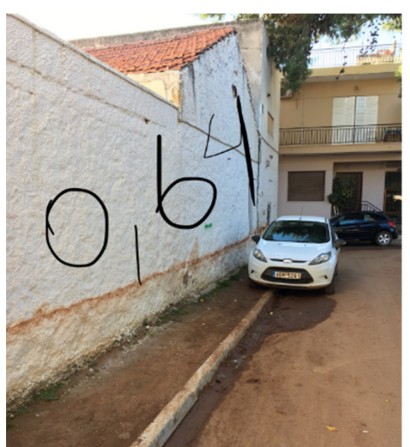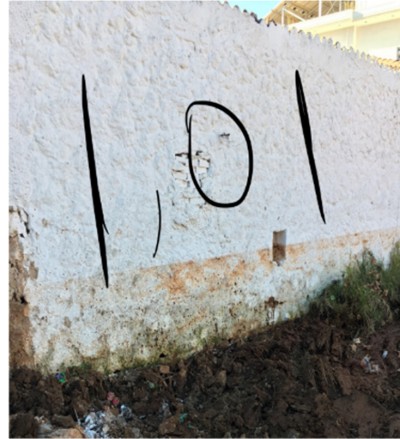

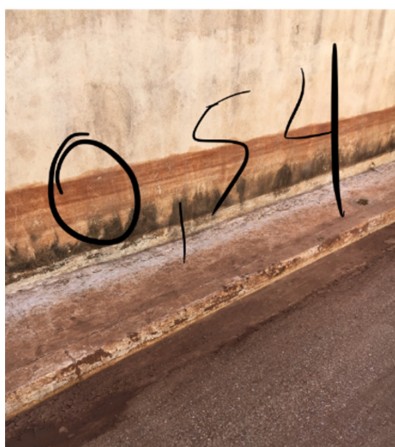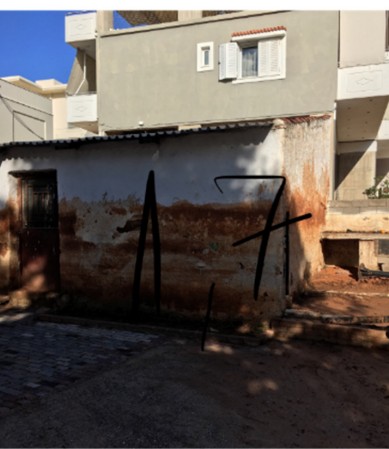

**Figure 4.** Indicative observation points in Mandra town (in roads): the observed maximum depths are 0.64 m (up & left), 1.01 m (up & right), 0.54 m (down & left), 1.70 (down & right).

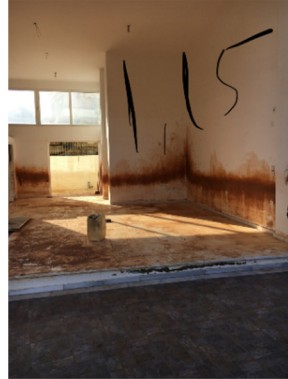
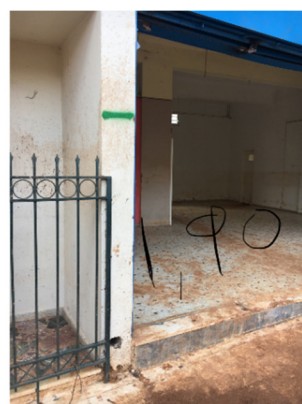
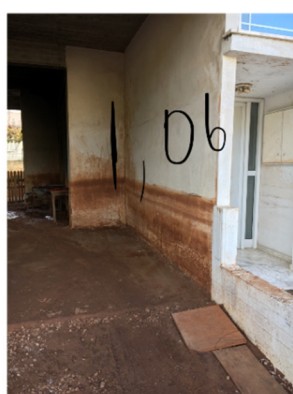

**Figure 5.** Indicative observation points in Mandra town (inside the buildings): the observed maximum depths are 1.15 m (left), 1.90 m (middle), 1.06 m (right).

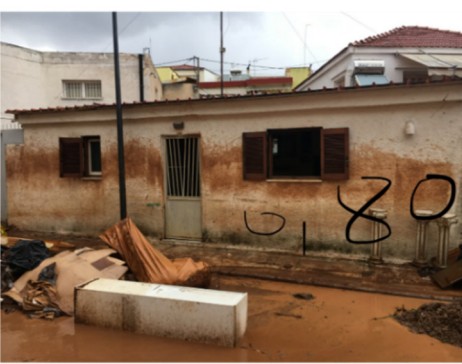
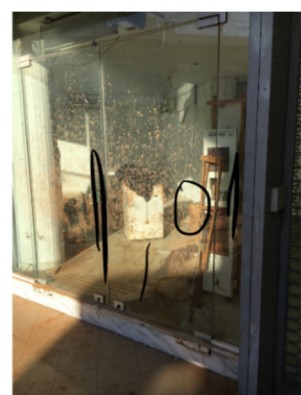

**Figure 6.** Uncertainty issues at the observation points: the finally selected values for the observed maximum depths are 0.80 m (left), 1.01 m (right).

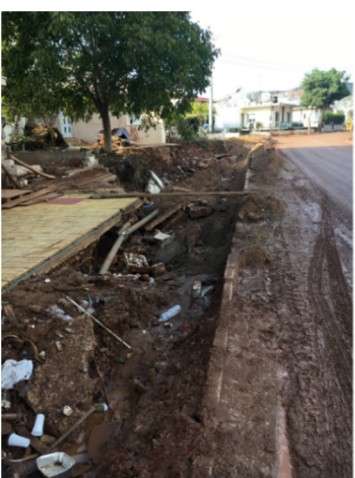
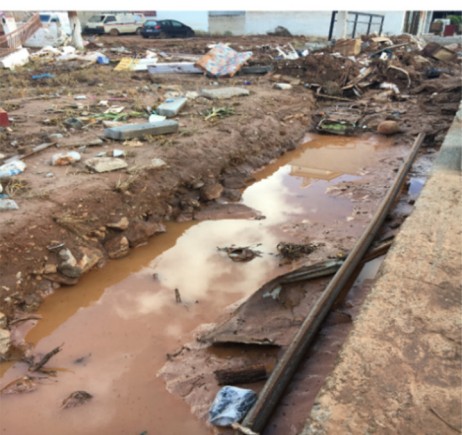

**Figure 7.** Example of the of the urban drainage system failure.

Specifically, Figure 4 is a representative photo of flood traces observed in the roads while Figure 5 is a representative photo of flood traces observed inside buildings. At some points, local inhabitants had marked with a green line the flood maxima. This information was used carefully, employing engineering judgement and after cross checking. It should be noted that the guidance of the locals in the field was substantial in order to address the uncertainties of the measurements and provided and increased reliability of the dataset. For example, the building in Figure 6 (left) has two mud traces: the lower trace denotes the real flood depth maximum, while the higher denotes mud splashing as a result of turbulence

from obstacles. The flood mark depicted in Figure 6 (right) is also another point which is characterized by uncertainties. All these issued were addressed doing forensic research and with the assistance provided by the local community in the form of small discussions, interviews, etc.

Finally, Figure 7 depicts an example of the sewer system failure. We should mention here that the main storm sewer system is underground and was designed for a maximum flood capacity of 10 m$^3$/s. The main pipe of the latter system is below the Agia Aikaterinin str. and its extensions (Vaggeli Koropouli str. and Str. Nik. Rokka str.) crossing the town. Based on the field visit and the interviews with the locals, it seems that the sewer system did not work (it was destroyed or filled with debris) and the flow inside the system was negligible compared to the flood wave that hit the town. For this reason, we did not include urban drainage flow in our hydrodynamic analysis.

### 2.3. Hydrodynamic Simulators

For the hydrodynamic simulations, we used two well-known 2D hydrodynamic models, namely the HEC-RAS [30] and the MIKE-FLOOD [31,32] pieces of software. Both simulators are based on the 2D Shallow Water Equations (2D-SWE), either in their full form or in the simpler diffusion wave form, while the equations are solved using the finite volume method. In this study, we exploited the 2D diffusion wave mode for both software since the full form of 2D-SWEs numerical instabilities were observed.

We used the same input data, parameters, and boundary conditions in both models. Specifically, the digital terrain model (DTM), the friction coefficients, the internal boundaries defined by the buildings, and the forcing 100 hydrographs as defined from the above-mentioned uncertainty analysis were introduced in both software. It should be noted that only the inflow of Agia Aikaterini catchment (and not Soures catchment inflow) was considered, since it was the only driver for the flood observed inside the town.

The computational area consists of a polygon of 1.6 km$^2$ area. The digital terrain model (DTM) which was used for geographical information was derived from the National Cadastre & Mapping Agency of Greece with a spatial resolution of 5 m. The internal boundaries for the buildings were manually designed with the assistance of satellite images in the Google Earth platform. For both pieces of software, the lateral boundaries of the computational domain were assumed to have a no-slip condition (solid boundaries) in order to preserve the water volume, while for the downstream boundaries the open boundaries mode was selected.

Regarding the HEC-RAS software, we manually inserted (233) flow breaklines around urban blocks, and then modified the mesh by generating orthogonal cells around the breaklines, thus improving the computational speed and accuracy of the model. For the whole mesh, 121,583 cells were generated with the average cell area being about 13 m$^2$. For the upstream boundaries, we used the mode of the input hydrograph (which requires no additional parameters), while for the downstream boundaries we selected the open boundaries, which are based on the Manning equation and require an extra parameter, namely the energy slope $S_f$. The time step was selected to be equal to 10 s.

The Manning coefficient of the computational domain $n_r$ was assumed to lie in a plausible range of values and was calibrated. Since HEC-RAS software is not capable of including internal solid boundaries, buildings are represented with a local increase of roughness, a common methodology in the relative literature [33]. Therefore, we used two more Manning coefficients, $n_h$ and $n_l$ (high and low), for dense and less dense urban blocks, respectively.

Regarding the MIKE software, we performed a similar procedure in order to generate the mesh, but with triangular elements. Specifically, we generated a mesh comprising 22.264 elements, with an average area of 70 m$^2$, while the smallest permitted triangular angle was 23°. For the upstream boundaries, we also used the mode of the input hydrograph, while for the downstream boundaries, we selected the open boundaries as well which required no additional parameters. A constant eddy viscosity coefficient was selected with

a value equal to 0.1, while the time step ranged between 0.001 and 0.002 s. Finally, the wet-dry threshold was given a value equal to 0.005 m.

In contrast to HEC-RAS, MIKE FLOOD has the option to represent the buildings with a free-slip boundary condition. Therefore, there was no need to introduce new parameters in order to simulate the flood in a built-up area. Figure 8 depicts the computational area, the building footprints, and the upstream/downstream boundaries, while Figure 9 depicts the mesh generation for both pieces of software in an indicative part of the town.

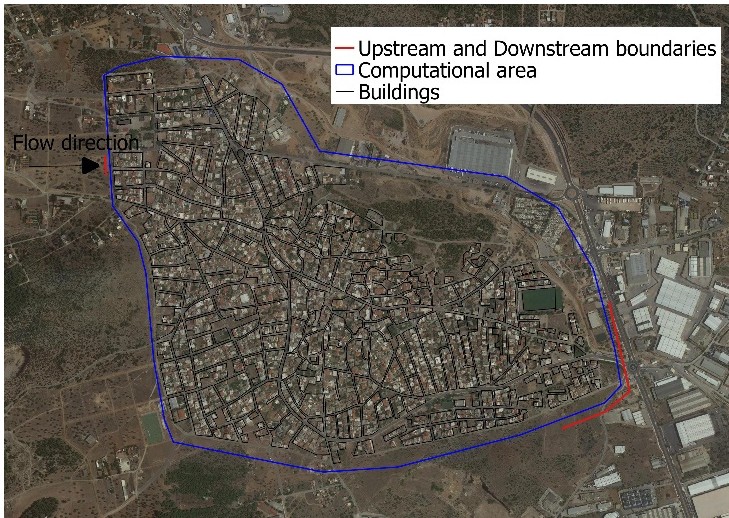

**Figure 8.** Computational area, upstream/downstream boundaries and buildings footprint used for HEC-RAS and MIKE FLOOD software.

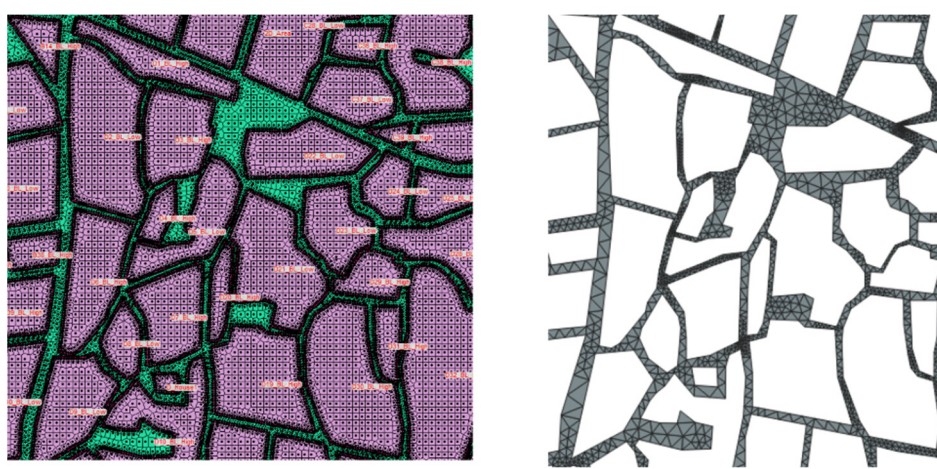

**Figure 9.** Mesh generation with HEC-RAS (left) and MIKE FLOOD (right) software.

*2.4. Calibration Strategy*

Although 2D hydrodynamic simulators have a strong physical base, they incorporate several grey-box parameters whose values, instead of being unique, adopted from handbooks or good practice guidance, lie in a plausible range. It is recommended that these parameters should be calibrated against relevant on-site measurements [14,34]. In theory, the latter simulators shall be calibrated against both water depths and flow velocities. However, the difficulty in finding observed velocities led us to limit the calibration only using maximum water depths. Moreover, the use of this kind of simulator for flood simulation has as a consequence an extremely high computational burden, which is exacerbated in cases where procedures, such as calibration, optimization, and uncertainty quantification,

are employed. For example, each HEC-RAS simulation required approximately 2 h while each MIKE simulation required 4 h in an Intel(R) i7-4790 CPU/3.60 GHz.

Overall, it is not necessary or feasible to incorporate all the parameters of a model in such processes if we aim at an efficient optimization. To this end, sensitivity analysis, a process aiming to screen the most influential parameters on model results, can help modelers and practitioners identify a model's key input parameters, and thus reduce computational cost.

The two most common classifications of sensitivity analysis methods are the global sensitivity analysis (GSA) and the local sensitivity analysis (LSA). It is not in the scope of the present work to describe in detail sensitivity analysis approaches and for further information the reader is referred to [35]. In recent literature, a wide variety of sensitivity analysis methods are presented (e.g., local methods, global methods, qualitative, quantitative methods) from different scientific fields [36,37]. Two of the most widely used sensitivity analysis methods are Sobol's sensitivity analysis method [38] and the screening technique proposed by [39]. The latter is one of the most widely used sensitivity analysis methods as it is easy to implement and is not time consuming [7,36].

Specifically, Morris [39] introduced the concept of elementary effects and proposed a sensitivity analysis method which is based on the computation of the mean and standard deviation of elementary effects in order to determine the effects of input parameter on the final result. According to [35], the input parameters may have: (i) negligible; (ii) linear and additive; or (iii) nonlinear or involved in interactions with other parameters. The mean of the elementary effects is used to assess the overall effect of the input parameter on the final result, while the standard deviation is used as a metric for the interactions with other parameters. However, [40] proposed to use the absolute mean instead of the mean in order to not introduce type II errors. In the present work, the sampling strategy proposed by [39] was used. Overall, the Morris elementary effects method can be categorized as a GSA approach, and it is simple, robust, and has low computational burden compared to other GSA methods (e.g., Sobol's method, GLUE etc.).

In the present study, we performed the calibration phase of the simulation only with HEC-RAS and we divided it in two stages: (a) First, we performed a GSA for the required model parameters using the SAFE toolbox [41] in order to reduce the dimensional space through parametric screening. Then, (b) we performed a grid-search calibration. The required five parameters were: a) the Manning coefficient for the roads, $n_r$; (b) the Manning coefficient for the urban blocks (high value), $n_H$; (c) the Manning coefficient for the urban blocks (low value) $n_L$; (d) the confidence interval of the uncertainty band of the upstream hydrograph, *CI*; and (e) the downstream energy slope, $S_f$. The range of values assigned to each of the five parameters are presented in Table 1.

**Table 1.** Parametric range.

| Parameter | Lower Limit | Upper Limit |
|---|---|---|
| $n_r$ [s/m$^{1/3}$] | 0.03 | 0.06 |
| $n_H$ [s/m$^{1/3}$] | 40 | 60 |
| $n_L$ [s/m$^{1/3}$] | 15 | 25 |
| *CI* [%] | 10 | 90 |
| $S_f$ [-] | 0.01 | 0.03 |

## 3. Results

### 3.1. Flood Dataset

Table 1 presents the maximum water depths measured at the 44 observation points, with their coordinates in both the Greek Geodetic Reference System 1987 (GGRS87) and the World Geodetic System 84 (WGS84).

In this link (https://zenodo.org/record/7140750, accessed on 3 October 2022), the reader can also download the full dataset on the Zenodo platform. The dataset consists of the following:

1. The maximum water depths recorded at the 44 points after the event with their coordinates in both GGRS87 and WGS84 systems, as shown in Table 2.
2. The digital terrain model (DTM) of the greater area with a resolution of 5 m, as provided by the National Cadastre and Mapping Agency of Greece.
3. A shape file with the boundaries of the computational area used for both HEC-RAS and MIKE FLOOD software, which coincide with the boundaries of the town of Mandra.
4. A shape file with the Mandra urban blocks footprint, as were manually drawn on Google Earth platform.
5. The ensemble of 100 hydrographs as depicted in Figure 2, which serve as the inflow from Agia Aikaterini catchment to Mandra town. They are derived by implementing the FLOW-R2D hydrodynamic simulator at the catchment scale, having as an input the rainfall field captured by the weather radar during the Mandra flood event [23].
6. The rainfall field of the greater area with a spatial resolution of 200 m and a temporal resolution of 2 min, recorded by the X-band weather radar of the National Observatory of Athens.

**Table 2.** Maximum water depths measured at the field.

| Gauge | X, Y (GGRS87) | φ, λ (WGS84) | Depth (m) |
|---|---|---|---|
| 1 | 455668.2588, 4214030.3444 | 38.075555095412, 23.496248974378 | 3.00 |
| 2 | 455714.0308, 4213859.4439 | 38.074017108644, 23.496781377525 | 1.41 |
| 3 | 455618.1325, 4214089.9909 | 38.076090202058, 23.495673789261 | 1.91 |
| 4 | 455648.7464, 4214045.8128 | 38.075693548981, 23.496025555384 | 2.60 |
| 5 | 455715.7245, 4213941.2321 | 38.074754298656, 23.496795637049 | 2.00 |
| 6 | 455714.7635, 4213840.7551 | 38.073848713598, 23.496790884875 | 2.40 |
| 7 | 455708.8934, 4213757.7429 | 38.073100288440, 23.496729087177 | 1.90 |
| 8 | 455718.1527, 4213696.6548 | 38.072550190317, 23.496838421089 | 2.60 |
| 9 | 455711.4275, 4213686.7258 | 38.072460378003, 23.496762362771 | 2.40 |
| 10 | 455693.3690, 4213585.9232 | 38.071551023452, 23.496562711634 | 3.30 |
| 11 | 455683.1461, 4213546.1162 | 38.071191767747, 23.496448625272 | 2.44 |
| 12 | 455694.2735, 4213521.9724 | 38.070974717633, 23.496576973349 | 2.00 |
| 13 | 455724.7554, 4213510.6998 | 38.070874612326, 23.496925173983 | 2.80 |
| 14 | 455815.0195, 4213431.7319 | 38.070167321865, 23.497959080654 | 2.43 |
| 15 | 455893.3008, 4213392.0692 | 38.069813674021, 23.498853945292 | 1.17 |
| 16 | 455950.5524, 4213396.1206 | 38.069852967810, 23.499506375731 | 1.28 |
| 17 | 456148.2251, 4213517.7609 | 38.070958813396, 23.501752448480 | 1.56 |
| 18 | 456149.9273, 4213543.6000 | 38.071191768069, 23.501770274694 | 0.88 |
| 19 | 456244.9914, 4213542.7794 | 38.071188961397, 23.502854094564 | 1.55 |
| 20 | 456349.9641, 4213582.0807 | 38.071548216307, 23.504048435197 | 1.19 |
| 21 | 456405.9467, 4213583.0278 | 38.071559443195, 23.504686605580 | 1.68 |
| 22 | 456498.3168, 4213604.9584 | 38.071761523636, 23.505738338489 | 3.00 |
| 23 | 456555.1662, 4213651.3694 | 38.072182522132, 23.506383639558 | 2.47 |
| 24 | 456597.3211, 4213639.6230 | 38.072078676069, 23.506864940258 | 1.19 |
| 25 | 456662.0088, 4213690.0413 | 38.072536158243, 23.507599370013 | 1.87 |
| 26 | 456690.6165, 4213659.3707 | 38.072261108064, 23.507927367552 | 1.06 |
| 27 | 455950.3622, 4213418.8560 | 38.070057859161, 23.499502811367 | 1.15 |
| 28 | 455722.5295, 4213734.9355 | 38.072895404700, 23.496885956174 | 2.00 |
| 29 | 455761.5685, 4213956.7620 | 38.074896496719, 23.497317344483 | 1.70 |
| 30 | 455764.4553, 4213970.1369 | 38.075017177319, 23.497349431868 | 1.30 |
| 31 | 455770.2635, 4213945.5041 | 38.074795460034, 23.497417170265 | 0.81 |
| 32 | 455783.0573, 4213940.4525 | 38.074750556643, 23.497563343334 | 1.10 |
| 33 | 455811.7063, 4213918.1875 | 38.074551291492, 23.497891341098 | 1.16 |
| 34 | 456228.1013, 4213834.3580 | 38.073815970735, 23.502643747389 | 1.01 |
| 35 | 456335.9814, 4213774.6124 | 38.073282717248, 23.503877306043 | 0.28 |
| 36 | 456306.8247, 4213760.7545 | 38.073156420710, 23.503545742317 | 0.61 |
| 37 | 456333.7110, 4213759.3652 | 38.073145194065, 23.503852350022 | 1.21 |

**Table 2.** *Cont.*

| Gauge | X, Y<br>(GGRS87) | φ, λ<br>(WGS84) | Depth (m) |
|---|---|---|---|
| 38 | 456346.2390, 4213821.5813 | 38.073706513488, 23.503991391677 | 1.01 |
| 39 | 456344.8884, 4213802.9037 | 38.073538118443, 23.503977130532 | 0.62 |
| 40 | 456422.4147, 4213797.8190 | 38.073496019748, 23.504861299648 | 0.54 |
| 41 | 456529.0244, 4213792.8917 | 38.073456726769, 23.506077031385 | 0.46 |
| 42 | 456528.4916, 4213751.4763 | 38.073083448814, 23.506073467053 | 0.54 |
| 43 | 456557.3220, 4213703.9877 | 38.072656842958, 23.506405030212 | 0.91 |
| 44 | 456652.5360, 4213731.8215 | 38.072912246041, 23.507488849352 | 0.78 |

*3.2. Calibration Phase*

As previously described, we first performed the GSA in order to reduce the number of parameters to be calibrated, and therefore, reduce the dimensional space. Based on engineering judgement and similar studies [23], we assumed that the trajectory number was equal to 15. Hence, the required number of HEC-RAS simulations was equal to 90.

For each simulation, we calculated the root mean square error (RMSE) between the simulated and the observed maximum water depths. Based on this value, the SAFE toolbox calculates the mean and the standard deviation of the elementary effects.

Figure 10 depicts the results of the sensitivity analysis. It seems that the *CI* is by far the most influential parameter regarding the RMSE. The impact of the forcing driver in the model output is in accordance with similar flood studies [42]. The second most influential parameter is the Manning coefficient of the computational domain $n_r$, while the remaining Manning coefficients and the energy slope required at the downstream boundaries seem to have a negligible impact on the RMSE.

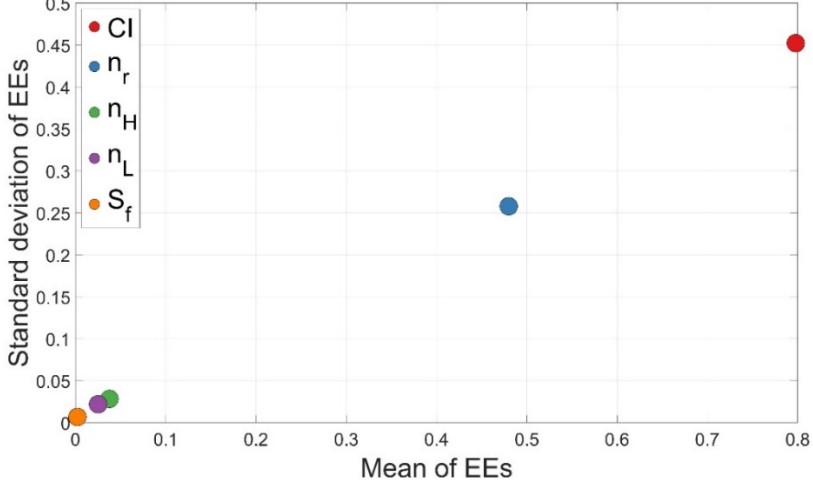

**Figure 10.** Sensitivity analysis results.

Based on the previous analysis, we then focused on the pair of the most influential parameters (*CI*, $n_r$). Therefore, we implemented a grid-search calibration, in order to find the optimal combination of the pair values. Assuming a step of 10% for the *CI* (which according to Table 1 ranges from 10% to 90%) and 0.01 s/m$^{1/3}$ for the $n_r$ (which according to Table 1 ranges from 0.03 s/m$^{1/3}$ to 0.06 s/m$^{1/3}$), we produced 9 × 4 = 36 scenarios with different combinations of *CI* and $n_r$, while the other parameters ($n_H$, $n_L$ and $S_f$) were assigned the average values 50 s/m$^{1/3}$, 20 s/m$^{1/3}$ and 0.02, respectively, according to Table 1.

Then, we performed again the hydrodynamic analysis for these scenarios, and we defined as an objective function the RMSE of the simulated maximum flood depths against the observed data. Figure 11 (up, left) depicts the dimensional space of the objective function. Since the target was the combination of the *CI* and $n_r$ values for which the RMSE

is minimized, it can be deduced that these values are 40% and 0.06 s/m$^{1/3}$, respectively (denoted by red star in the figure), while the RMSE value is equal to 0.70 m.

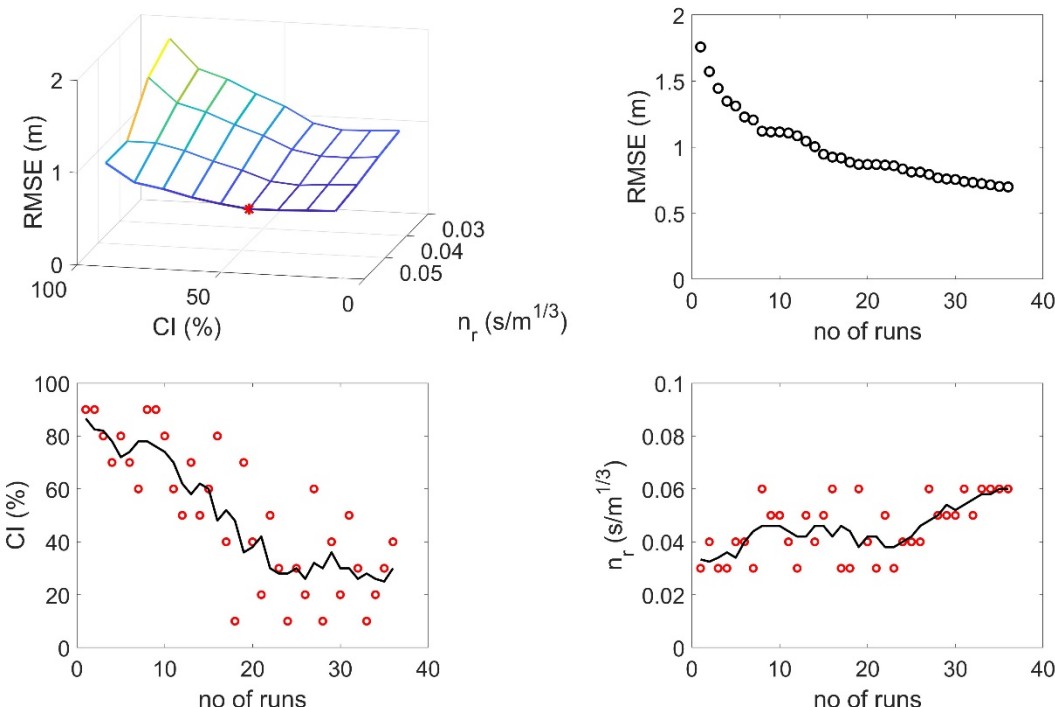

**Figure 11.** Dimensional space for the optimization function used for the grid-search calibration (up & left); Runs are sorted from maximum to minimum RMSE (up & right); Confidence Interval *CI* in respect to the run number with 5-period moving average denoted by the black line (down & left); Manning coefficient $n_r$ in respect to the run number with 5-period moving average denoted by the black line (down & right).

It should be noted that the optimization procedure is trapped in the boundaries, which means that the dimensional space is not sufficiently explored. If the runs are sorted from the maximum until the minimum RMSE, it can be observed (Figure 11, up, right) that the objective function can be further minimized, but not that much. Taking into account the computational cost, we did not perform additional runs. Finally, Figure 11 (down, left and down, right) depicts the general trend of the calibrated parameters *CI* and $n_r$ in respect of the run number, as previously sorted. It seems that the parameters tend to reach their optimal values as the calibration is in progress. This is not a proof that we achieved the global optimum, but it is a strong indication that we avoided the equifinality issue.

### 3.3. Calibrated HEC-RAS vs. Informed MIKE FLOOD

Since we assumed that we identified the optimal combination for the HEC-RAS parameter values, we then tried to identify whether we can inform MIKE FLOOD with these values. With this blind test, we aimed to investigate whether these calibrated values have a global nature or are model-specific.

As previously described, MIKE FLOOD includes the option for representing the buildings using a free-slip condition, which in general is better practice than the other two available methodologies in the literature, namely the local increase of the elevation or of the Manning coefficient [33]. Therefore, there is no need for estimating the Manning coefficient for the urban blocks. Besides, there is also no need for estimating a parameter for the downstream open boundaries, in contrast with HEC-RAS which requires the energy slope $S_f$. In order to make a meaningful comparison, MIKE FLOOD configuration (computational area, boundary conditions, bathymetry, boundaries of buildings) was the same as in HEC-RAS. Furthermore, for the MIKE-FLOOD simulations, the forcing driver was the

hydrograph with the optimal value of *CI* = 40%, as described in the previous section, while the Manning coefficient of the computational domain $n_r$ was given the optimal value of 0.06 s/m$^{1/3}$. It was found that the RMSE metric was significantly bigger than the corresponding RMSE derived by HEC-RAS, which was equal to 1.63 m

Figure 12 depicts the comparison of the observed maximum flood depths vs. the simulated results derived from HEC-RAS and MIKE FLOOD. The inundation maps shown in Figures 13 and 14 are derived with the calibrated HEC-RAS and the informed MIKE FLOOD, respectively, and depict the maximum water depths. Figure 15 depicts the differences between the calibrated HEC-RAS and the informed MIKE FLOOD, for maximum water depths, while Figure 16 depicts the distribution of the residuals.

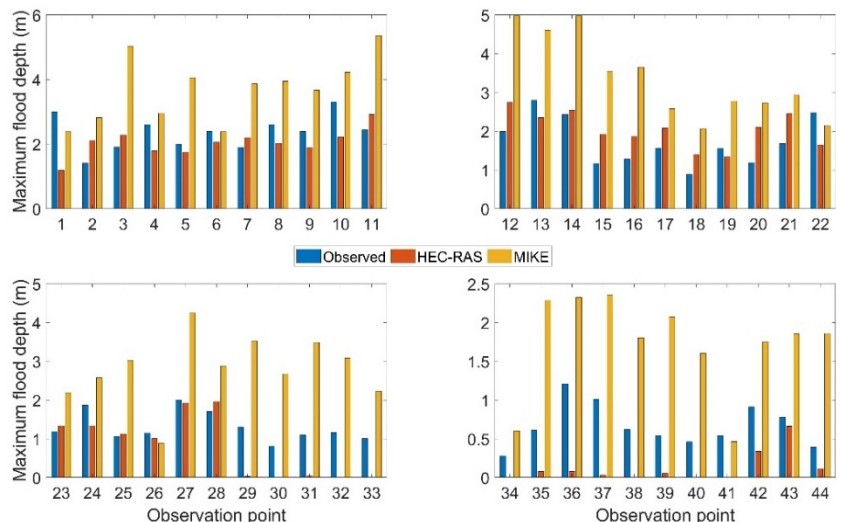

**Figure 12.** Comparison of the maximum flood depths derived by field observations: HEC-RAS simulation and MIKE FLOOD simulation.

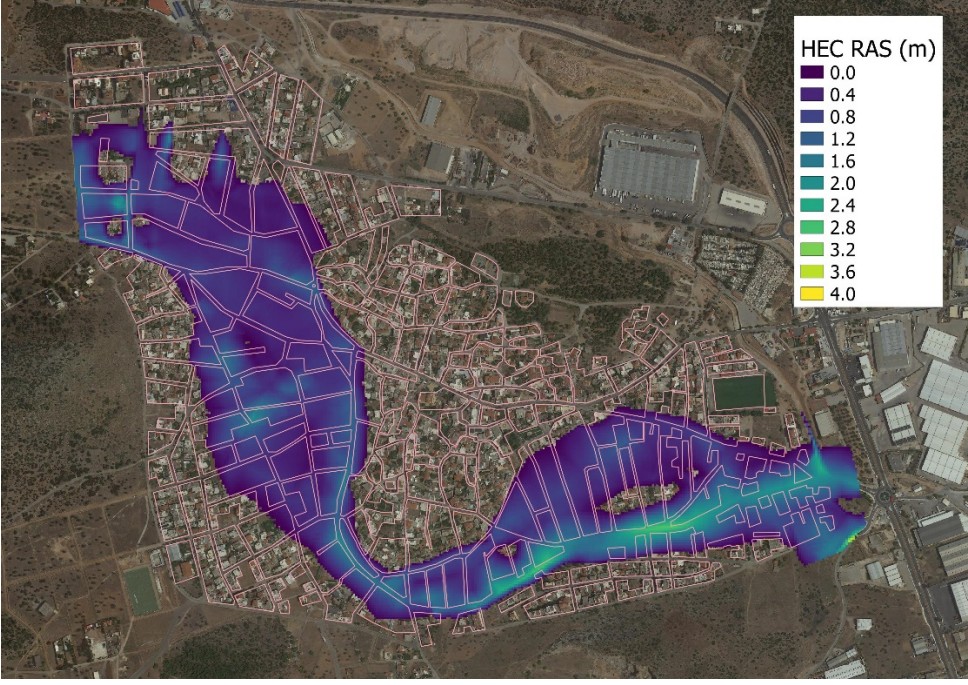

**Figure 13.** Results of the calibrated HEC-RAS for maximum water depths.

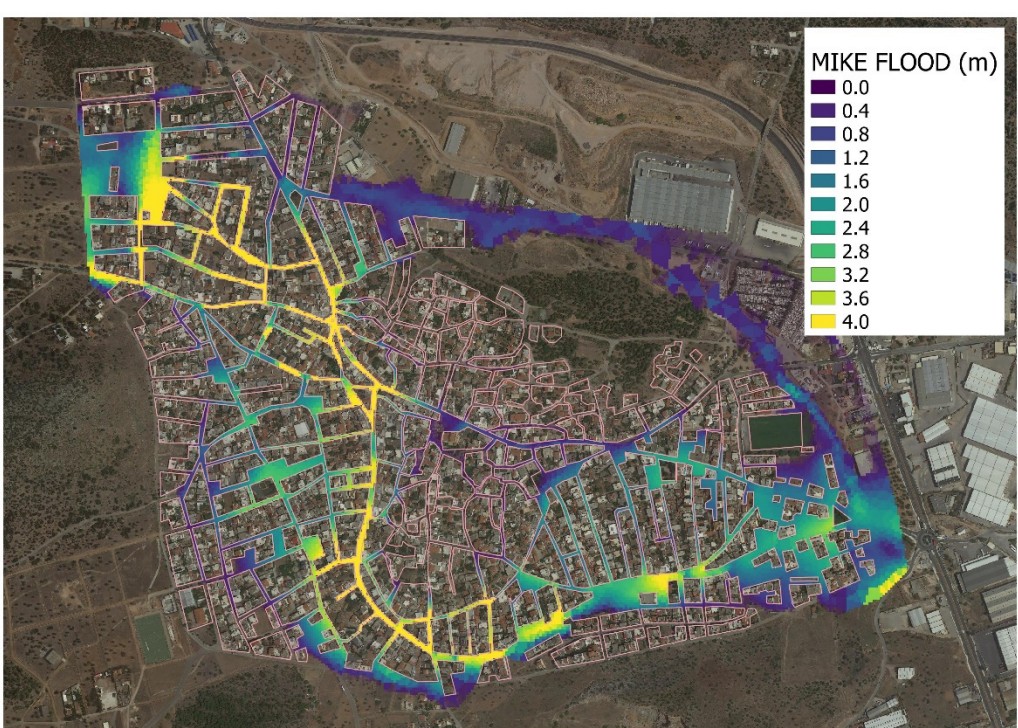

**Figure 14.** Results of the informed MIKE FLOOD for maximum water depths.

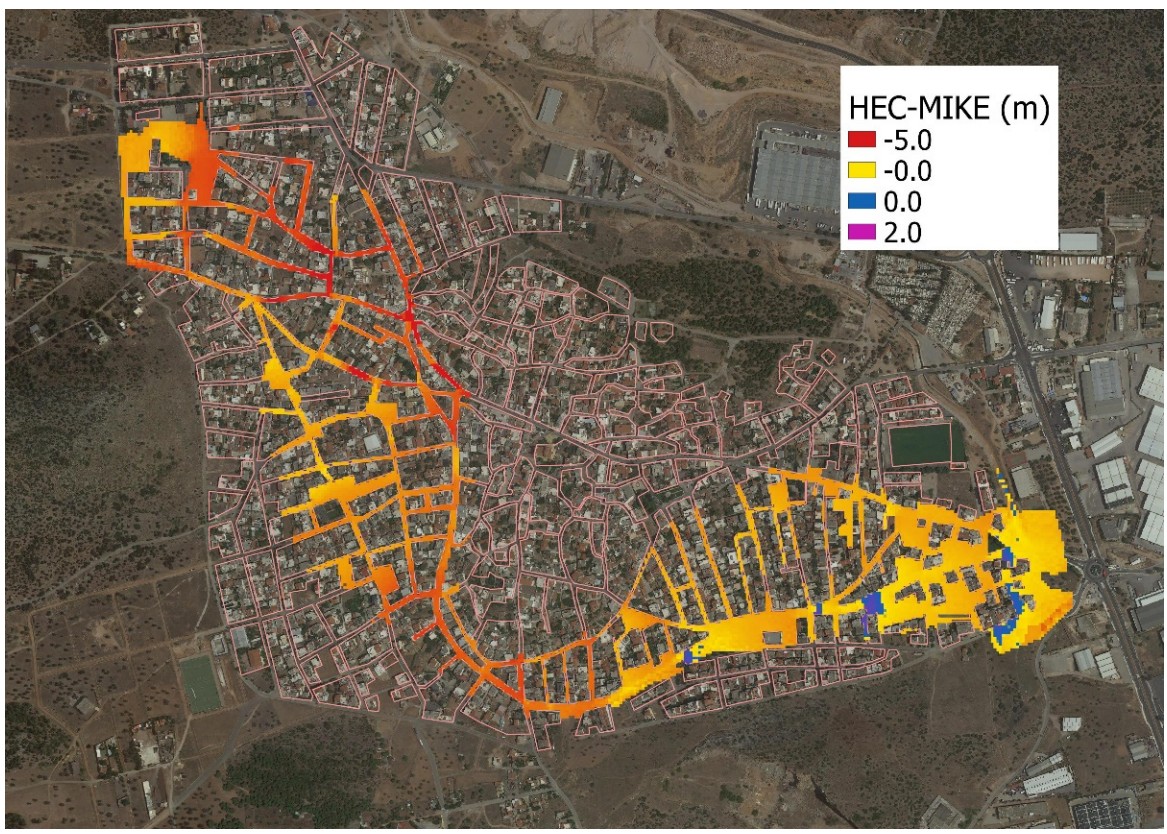

**Figure 15.** Calibrated HEC-RAS minus informed MIKE FLOOD for maximum water depths.

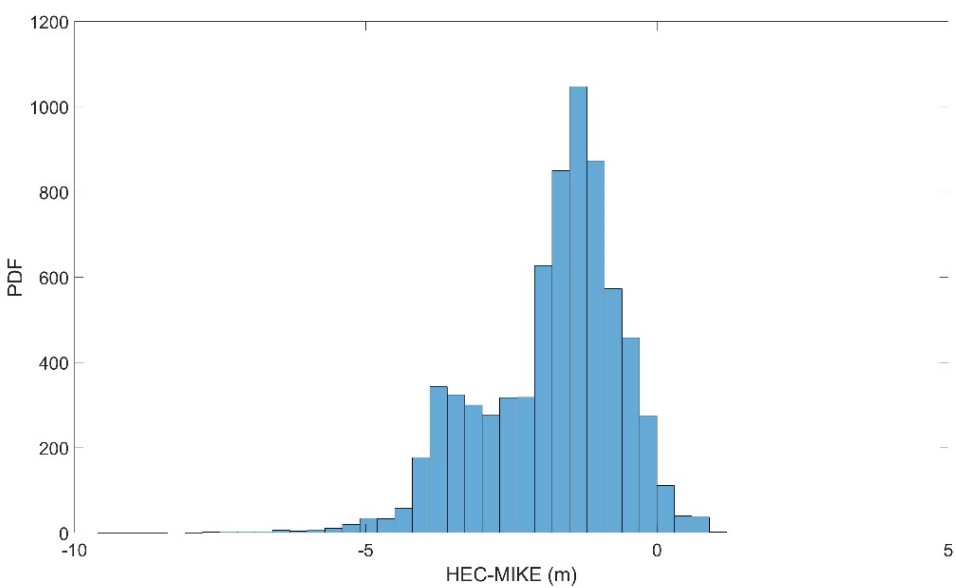

**Figure 16.** Distribution of the residuals HEC minus MIKE.

There is an area of the city where both models failed to reproduce the flood sufficiently, either overestimating or underestimating the maximum observed flood depths (gauges 29–41). This is probably due to each model's structure and the simplifying assumptions regarding the validity of the diffusion wave model. For the remaining observation points, the calibrated HEC-RAS seems to capture the flood characteristics reasonably well. On the other hand, the informed MIKE FLOOD seems to overestimate flood depths in a systematic way. The residual (HEC-RAS minus MIKE FLOOD results) shows that an overestimation of flood depths by MIKE FLOOD compared to the ones by HEC-RAS occurs in the upstream and the main computational area, while in some parts of the downstream computational area we observe the opposite. The residual distribution ranges from –5 m (MIKE FLOOD is overestimating compared to HEC-RAS) to 1 m (HEC-RAS is overestimating compared to MIKE FLOOD). It should be noted that the abstraction is performed just for the intersection between these two flood extents.

## 4. Discussion and Concluding Remarks

Our work had a dual objective: First, to perform forensic research, collecting post-event flood data in the field, and second, to investigate several modelling issues regarding the use of well-known flood simulators in this kind of complex case studies. More specifically, in this paper we presented and shared a flood dataset for the Mandra flood event, which occurred in the greater metropolitan area of Athens, Greece, on 15 November 2017. This real-world flood dataset was used to calibrate flood simulations by the HEC-RAS software and used to inform the MIKE FLOOD software. The open flood dataset should be welcome given the scarcity of this kind of data. Since many simulators are usually verified with numerical experiments and analytical or simplified physical models, this dataset can potentially contribute to benchmarking robust flood simulators, tested in real world case studies.

The major findings of our work consist of: (a) the lessons learned from a post-event data collection in the field; (b) the calibration issues raised from a computationally demanding simulator; (c) the answer to the research question regarding the global use of the calibrated parameters.

As far as the first finding is concerned, it seems that a post-event forensic research is feasible, in terms of resources and equipment. Since urban flooding is characterized by mud flows, there are lot of clues for the maximum water depth observed in several points. The drawback is the absence of data indicating the time evolution of the phenomenon and flow velocity data. The participation of the locals in the field survey was of high importance

in order to derive a reliable and representative flood dataset. Small interviews and chats could provide answers for several unclear points.

Regarding the second finding, the proposed two-phase calibration procedure, including the parametric screening and the grid-search methodology, seems to be an efficient way of reducing the computational demand of the simulator. Although the calibration procedure seems to be trapped in the boundaries of the dimensional space, there are strong indications that the optimized pair of parameters gives the optimal result for these model structures, without equifinality problems.

Finally, the concept of the informed modeling does not seem to work. Possible reasons for this are the differences regarding the way in which buildings are represented, as well as the different form of the downstream open boundaries and the mesh structure. However, we strongly highlight that the systematic overestimation of MIKE FLOOD against the observed data does not indicate that one software is better than the other, but that every software has its dynamics, and the transferability of parameter values cannot be performed in blind trust, while the direct calibration of model input parameters is a must. This reinforces the belief that the flood model parameters are of grey-box nature and their global use should be avoided or adopted with the utmost care.

**Author Contributions:** Conceptualization, V.B., I.K. and V.A.T.; methodology, V.B. and I.K.; software, E.R. and S.H.; validation, V.B. and I.K.; investigation, V.B., I.K., E.R. and S.H.; data curation V.B. and J.K.; writing—original draft preparation, V.B. and I.K.; writing—review and editing, J.K., I.A.S. and V.A.T.; visualization, V.B., I.K. and S.H.; supervision, V.B., I.A.S. and V.A.T. All authors have read and agreed to the published version of the manuscript.

**Funding:** This research received no external funding.

**Data Availability Statement:** The data presented in this study are openly available in Zenodo at https://doi.org/10.5281/zenodo.7140750 (accessed on 3 October 2022).

**Acknowledgments:** We thank Moussoulis of DHI in Athens for the educational license of MIKE FLOOD.

**Conflicts of Interest:** The authors declare no conflict of interest.

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
