# Peer review of "Identifying Modelling Issues through the Use of an Open Real-World Flood Dataset"

_hydrology, doi:10.3390/hydrology9110194_

Round 1

Reviewer 1 Report

The paper is interesting in two ways; 1. It collects field data right after major flooding in a town, and 2. It uses the filed data to calibrate and validate the numerical models of HECRAS and MIKEFLOOD. During the calibration, it employs one of the widely used optimization methods. The paper has a contribution to the literature, and therefore it can be accepted after minor revision.

1.      The title can be revised since, as it is, it sounds like a review paper. It may be revised as ‘Modelling real flood event using field data and numerical model; Case study: Mandra Town, Greece’

2.      As seen in Figure 3, the observation locations are chosen along a road (I guess), not scattered over the town; why?

3.      While referring to the pictures in Figures 4, 5, and 6 within the text, it would be helpful to mention the measured max water levels so that a reader can have a feeling!

4.      In numerical modeling, the lateral boundaries are assumed to be solid. Do you mean ‘no flow condition’?

5.      In Table 1, what is ‘the interval coefficient’? It would be helpful to expand this term within the text.

6.      There should be a Conclusions section in which you need to highlight the major findings.

7.      In the Discussion section, discuss the application and numerical outcomes in light of related existing literature. The overestimation/underestimation problem, handling of the buildings and the boundary conditions, and the studies of Haltas et al. 2016 (Water Resources Management and Natural Hazards). In those studies, the handling of the especially buildings is interesting.

‘Two-dimensional numerical modeling of flood wave propagation in an urban area due to Urkmez dam-break, Izmir, Turkey’ By: Haltas, Ismail; Tayfur, Gokmen; Elci, Sebnem, NATURAL HAZARDS   Volume: 81   Issue: 3   Pages: 2103-2119   Published: APR 2016

Testa, G., Zuccala, D., Alcrudo, F., Mulet, J., and Frazao, S. S. (2007). “Flash flow experiment in a simplified urban district.” Journal of Hydraulic Research, 45, 37–44.
Tingsanchali, T. and Chinnarasri C. (2001). ‘Numerical modelling of dam failure due to flow overtopping’, Hydrological Sciences Journal, 46(1), 113-130, DOI: 10.1080/02626660109492804

Author Response

In this document we try to address all the issues raised by the reviewer #1. Our reply is with the red font while the manuscript quotation with blue font.

Reviewer 2 Report

Authors present an interesting approach for a practical engineering problem related to urban flood management. It is focus on the calibration process of hydraulic models applied to floodplain simulation.

It is well structured and presented. However, some aspects should be deppened to help the readers to understand the  the novelty and improvement it brings.

Authors observed that MIKE model tends to overestimate, but the fact that the HECRAS model tends to profoundly underestimate is not sufficiently discussed. It would be opportune to evaluate the impact of these differences on the flood risk analysis. It is, if this level of accuracy achieved on the calibration process result enough for urban flood management purposes.

Only water depth are considered in the assessment, however, water velocity is also relevant to evaluate flood hazard.

Regarding the Discussion Section, it is not clear what are the final conclusions and recommendations that authors want to convey.

These suggestions aim to improve the utility of this interesting research paper.

Author Response

In this document we try to address all the issues raised by the reviewer #2. Our reply is with the red font while the manuscript quotation with blue font.
